# The impact of background liver disease on the long-term prognosis of very-early-stage HCC after ablation therapy

**Kenta Takaura**[1], **Masayuki Kurosaki**[1], **Kento Inada**[1], **Sakura Kirino**[1], **Kouji Yamashita**[1], **Tomohiro Muto**[1], **Leona Osawa**[1], **Shuhei Sekiguchi**[1], **Yuka Hayakawa**[1], **Mayu Higuchi**[1], **Shun Kaneko**[1], **Chiaki Maeyashiki**[1], **Nobuharu Tamaki**[1,2], **Yutaka Yasui**[1], **Jun Itakura**[1], **Kaoru Tsuchiya**[1], **Hiroyuki Nakanishi**[1], **Yuka Takahashi**[1], **Namiki Izumi**[1] *

1 Department of Gastroenterology and Hepatology, Musashino Red Cross Hospital, Tokyo, Japan,
2 Division of Medicine, NAFLD Research Center, University of California, San Diego, La Jolla, California, United States of America

* izumi012@musashino.jrc.or.jp

## Abstract

### Background and aim

The long-term prognosis of hepatocellular carcinoma (HCC) treated at a very-early-stage (the Barcelona Clinical Liver Cancer (BCLC) classification stage 0) was unclear, especially in terms of background liver disease.

### Methods

This single-center, retrospective study included 302 patients with BCLC stage 0 HCC treated with radiofrequency ablation (RFA) and followed for at least six months. We examined the impact of background liver disease on overall survival and recurrence.

### Results

The median age was 72 (range; 36–91) years; the median tumor diameter was 15 (range; 8–20) mm. The etiologies of background liver disease were hepatitis B virus infection (HBV) in 24 cases, hepatitis C virus infection (HCV) in 195 cases, and non-viral (NBNC) in 83 cases. Among the patients with HCV, 63 had achieved sustained virological response (SVR) by antiviral therapy (HCV SVR) before developing HCC (n = 37) or after HCC treatment (n = 26), and 132 had active HCV infection (HCV non-SVR). The median overall survival was 85 (95% CI; 72–98) months, and the median recurrence-free survival was 26 (95% CI; 20–30) months. Active infection with hepatitis C virus negatively contributed to overall survival (HR 2.91, 95% CI 1.31–3.60, p = 0.003) and recurrence-free survival (HR 1.47, 95% CI 1.06–2.05, p = 0.011).

### Conclusions

The prognosis of RFA treatment for very early-stage HCC was favorable. Achieving SVR in hepatitis C was important for further prognosis improvement.

**Data Availability Statement:** All relevant data are within the paper and its Supporting Information files.

**Funding:** Masayuki Kurosaki (MK) received funding support from the Japan Agency for Medical Research and Development (grant number: JP20fk0210067h0001), and Namiki Izumi (NI) received funding support from Japanese Ministry of Health, Welfare and Labor (H3-Kansei-Shitei-003).

**Competing interests:** I have read the journal's policy and the authors of this manuscript have the following competing interests: Namiki Izumi, Masayuki Kurosaki, and Kaoru Tsuchiya received lecture fees from Eisai, Bayer, Lilly, and Chugai. This does not alter our adherence to PLOS ONE policies on sharing data and materials.

## Introduction

Hepatocellular carcinoma (HCC) remains the fourth leading cause of cancer-related death globally, the second leading cause of death among men, and the sixth leading cause of death among women [1,2]. The incidence of HCC in East Asia, including Japan, is still high at 26.8/100,000 males and 8.7/100,000 females [3].

The majority of HCCs have a chronic liver disease background, with a high proportion of those with viral hepatitis caused by hepatitis B virus (HBV) or hepatitis C virus (HCV) [4]. For HBV, nucleoside/nucleotide analogs (NA) can be used to control virus replication [5–9]. For hepatitis C, direct-acting antivirals (DAAs) have recently made it possible to achieve a high rate of sustained viral response (SVR) [10–13], both of which could improve the prognosis of patients with chronic viral liver disease [14].

In the guidelines, treatment options for early-stage HCC within the Milan criteria include radiofrequency ablation (RFA), hepatic resection, and liver transplantation [15–17]. The RFA and hepatic resection remain the most common options in Japan. As for treatment outcomes, the results of the interim analysis of the SURF trial comparing RFA and hepatic resection indicate that the recurrence-free survival for HCC within the Milan criteria is equivalent [18].

On the other hand, although there are scattered reports on the prognosis after RFA for early-stage HCC [19], the long-term prognosis of HCC that falls into the very early-stage (stage 0) of the Barcelona Clinical Liver Cancer (BCLC) classification, which is single and less than 2 cm, is unclear.

In this study, we investigated the long-term prognosis of very early-stage HCC after RFA and analyzed the impact of background liver disease on survival and recurrence.

## Methods

This retrospective study examined patients who underwent RFA for HCC at the Musashino Red Cross Hospital between April 1999 and June 2020. The study included patients with a single HCC less than 2 cm (BCLC stage 0) who underwent RFA as initial treatment and follow-up of more than six months. Consecutive patients treated with RFA were included in the present study. Therefore, there were no exclusion criteria for this study.

The HCC diagnosis was made histologically or via imaging with contrast-enhanced computer tomography (CT), gadolinium ethoxybenzyl diethylenetriamine pentaacetic acid-enhanced magnetic resonance image (EOB-MRI), or contrast-enhanced ultrasonography. The efficacy of RFA treatment was determined within one week after treatment using CT or MRI according to mRECIST [20].

All RFA procedures were performed percutaneously under local anesthesia and transvenous analgesic administration. Real-time ultrasound (US) was chosen as the guidance modality. We used artificial ascites for tumors, which were in the hepatic dome or adjacent to the gastrointestinal tract. When performing RFA, we use a guiding needle with an external insulated sheath and one of the following devices: VIVA$^{TM}$ RF system (STARmed®); Cool-tip$^{TM}$ RFA system (Covidien®). The device was chosen based on the operator's decision. Ultrasonography was performed with a 3.0- to 6.0-MHz convex probe and the Aloka SSD-5500 (Aloka, Tokyo, Japan), Sonoline Elegra (Siemens, Erlangen, Germany), Aplio XV, XG and 500 systems (Toshiba Medical Systems-Canon Medical Systems, Tokyo, Japan). The degree of safety margin for RFA was set at 5mm. In this study, the clinical success rate of the initial RFA procedure was 92.7%. Treatment complications were seen in 2.3% and there were no treatment-related deaths. An obvious residual lesion on imaging, additional RFA, trans-arterial chemo-embolization (TACE), hepatic resection, or stereotactic radiotherapy was performed

after considering the localization of the disease, liver function, and the presence of comorbidities.

Time zero for the follow-up was the first RFA treatment, and survival and recurrence were examined. The patients underwent imaging surveillance to check for recurrence after RFA every 3–6 months by contrast-enhanced CT, EOB-MRI, contrast-enhanced ultrasonography, or ultrasonography. Although tumor markers, such as alpha-fetoprotein (AFP) and des-γ-carboxy prothrombin (DCP), were used as adjunctive diagnostics in the surveillance, the definitive diagnosis was made by imaging.

The etiologies of background liver disease were defined as HBV in HBs antigen-positive patients, HCV in HCV antibody-positive patients, and non-viral (NBNC) in both negative patients. The baseline data at RFA used in the study were age, gender, aspartate aminotransferase (AST), alanine aminotransferase (ALT), platelet count (PLT), prothrombin time (PT), total bilirubin (T.Bil), albumin (Alb), AFP, DCP, maximal tumor diameter, and etiology of liver disease. For PT and DCP, patients taking warfarin or direct oral anticoagulant (DOAC) were treated as missing values. As an evaluation of liver function, the albumin-bilirubin (ALBI) score was calculated using the following formula [21].

$$\text{ALBI score} = (\log 10 \text{ bilirubin } (\mu \text{mol/L}) \times 0.66) + (\text{albumin } (\text{g/L}) \times -0.085)$$

Based on the ALBI score, the patients were classified into four groups according to the mALBI grade: mALBI grade 1: −2.60 or less, 2a: −2.60 or more and less than −2.27, 2b: −2.27 or more and less than −1.39, and 3: −1.39 or more. 2b: −2.27 or higher and −1.39 or lower, and 3: −1.39 or higher [22].

Written informed consent was obtained from all patients before the RFA procedure. The requirement for written informed consent to be included in the study was waived because of the retrospective design of the study. The privacy of all patients was fully protected. Information about this study was made available to the patients (posted on the hospital website), and the research subjects were given the opportunity to opt out. The study protocol conformed to the ethical guidelines of the Declaration of Helsinki. The study was approved by the institutional ethics review committee of the Musashino Red Cross Hospital (ethical approval number: 636).

## Statistical analysis

Overall survival and recurrence-free survival after HCC treatment were examined using the Kaplan-Meier method. The difference in survival and recurrence-free survival between the two groups was examined using the log-rank test. Factors associated with survival and recurrence were examined using the Cox proportional hazards model. Values of p < 0.05 were considered statistically significant. Statistical analysis was performed using EZR.

## Results

### Baseline characteristics

The background of 302 patients is shown in Table 1. The median age at first RFA treatment was 72 (range; 36–91) years, the median tumor diameter was 15 (8–19) mm, and the etiologies were HBV in 24 (7.9%) cases, HCV in 195 (64.6%) cases, and NBNC in 83 (27.5%) cases.

The NA therapy was introduced in all HBV patients either before HCC development, at HCC diagnosis, or after the RFA treatment for HCC. In patients with HCV, 63 had achieved sustained virological response (SVR) by antiviral therapy (HCV SVR) either by interferon (IFN) regimen or IFN-free regimen, and 132 had active HCV infection

**Table 1. Baseline characteristics.**

| Variable | |
|---|---|
| Age(years old) | 72(36–91) |
| Sex(male/female) | 151/151 |
| AST(IU/L) | 41(14–282) |
| ALT(IU/L) | 34(9–245) |
| PLT($10^4$/μL) | 11.5(3.3–92.3) |
| PT(%) | 94(53–138) |
| T.Bil(mg/dL) | 0.8(0.2–3.1) |
| Alb(g/dL) | 3.8(2.5–4.9) |
| ALBI score | −2.45(−1.13–3.38) |
| AFP(ng/mL) | 12(1.5–3560) |
| DCP(mAU / mL) | 21(7–1110) |
| Tumor size(mm) | 15(8–19) |
| etiology | |
| HBs Ag positive | 24 |
| HCV Ab positive | 195 |
| SVR achived before RFA | 37 |
| SVR achived after RFA | 26 |
| both negative | 83 |

AST, aspartate aminotransferase; ALT, alanine aminotransferase; PLT, platelets; PT, prothrombin time; T.Bil, total bilirubin; Alb, albumin; ALBI, albumin–bilirubin; AFP, alpha-fetoprotein; DCP, des- γ -carboxy-prothrombin; HBs Ag, hepatitis B virus surface antigen; HCV Ab, hepatitis C virus antibody; SVR, sustained viral response; RFA, radiofrequency ablation.

throughout the observation (HCV non-SVR). We attempted to achieve SVR with an IFN regimen or IFN-free regimen at diagnosing HCV chronic infection or after confirming that there was no HCC recurrence for three months after RFA. Among the patients in the HCV SVR group, SVR was achieved before HCC development in 37 cases and after HCC treatment in 26 cases.

## Overall survival

Fig 1A shows survival curves for all patients. The median observation period was 42 (6–158) months, the median overall survival was 85 (95% confidence interval [CI]; 72–98) months, and the survival rates at 3, 5, 7, and 10 years were 88.0%, 70.8%, 51.1%, and 33.2%, respectively. The median overall survival by background liver disease was 98 (95% CI; 46–NA) months for HBV, 73 (95% CI; 66–91) months for HCV, and 145 (95% CI; 86–NA) months for NBNC (p = 0.12) (Fig 1B).

## Recurrence-free survival

Similarly, Fig 2A shows the recurrence-free survival curve for all patients. The median recurrence-free survival was 26 (95% CI; 20–30) months, and the recurrence-free survival rates at 1, 3, and 5 years were 72.4%, 35.7%, and 22.7%, respectively. Median recurrence-free survival by background liver disease was 23 (95% CI; 11–39) months for HBV, 26 (95% CI; 20–30) months for HCV, and 30 (95% CI; 17–38) months for NBNC (p = 0.48) (Fig 2B).

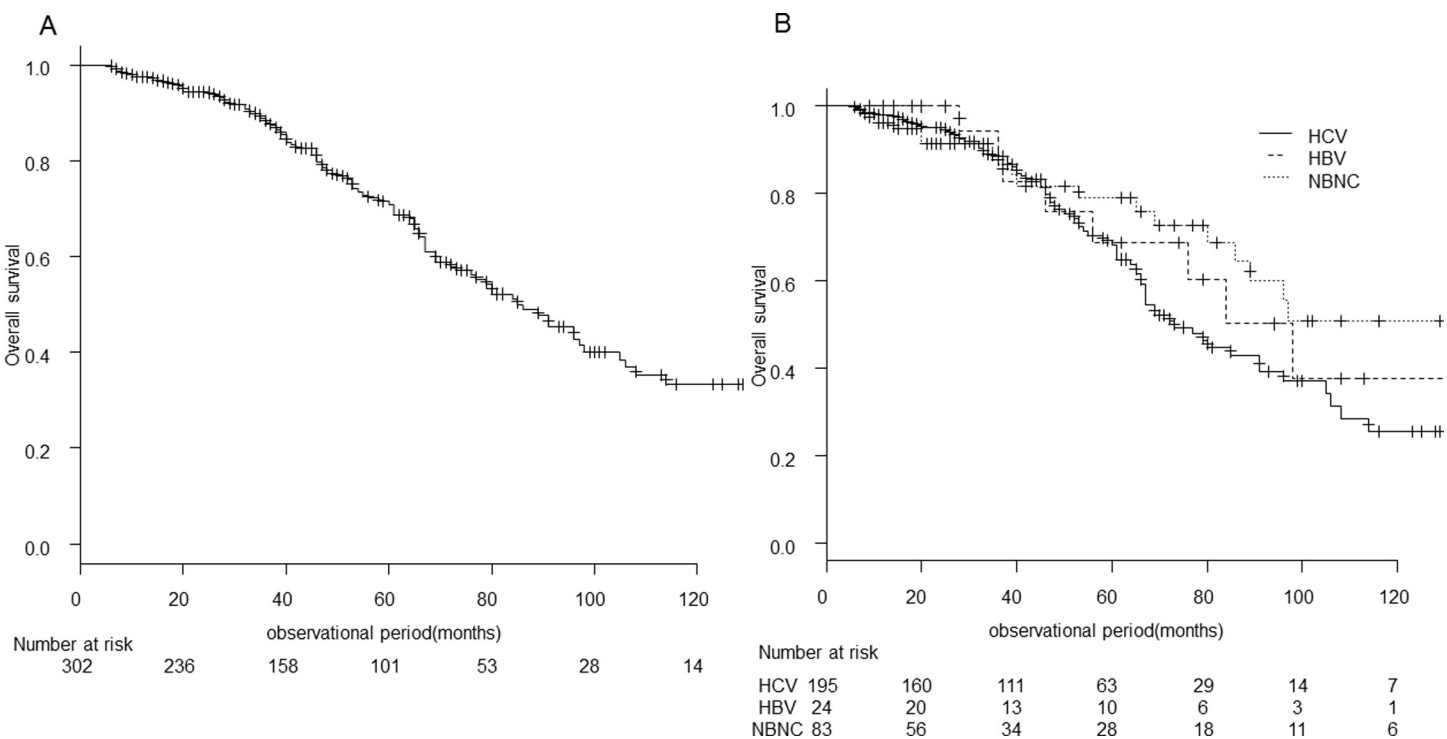

**Fig 1. Overall survival of very-early-stage HCC undergoing RFA.** The overall survival curve for all patients before (A) and after stratification by background liver disease (B).

### Factors related to overall survival

Table 2 presents the results of univariate and multivariate analyses of factors contributing to overall survival. Risk factors contributing to overall survival were HCV non-SVR (hazard ratio [HR] 2.91, 95% CI 1.31–3.60, p = 0.003).

The overall survival was compared between patients with active HCV infection during the observation period (HCV non-SVR) and others, including HBV, HCV SVR, and NBNC (Fig 3). Patients with active HCV infection had a significantly shorter median overall survival of 66 (95% CI, 58–73) months compared with 145 (95% CI, 96-not reached) months in the others (p < 0.001). Survival rates at 3, 5, 7, and 10 years were 84.7%, 59.4%, 33.3%, and 12.8% in patients with active HCV infection versus 91.0%, 82.8%, 71.5%, and 56.5% in the others.

### Factors related to recurrence-free survival

Table 3 shows the results of univariate and multivariate analyses of factors contributing to recurrence. Risk factors contributing to recurrence were tumor diameter ≥15 mm (HR 1.50, 95% CI 1.10–2.01, p = 0.011) and HCV non-SVR (HR 1.47, 95% CI 1.06–2.05, p = 0.011).

When we examined recurrence-free survival by these two risk factors (Figs 4 and 5), the median recurrence-free survival was significantly shorter in patients with larger tumors (the tumor diameter of 15–20 mm) than those with the tumor diameter of less than 15 mm: 20 (95% CI, 17–28) months versus 34 (95% CI 24–45) months (p = 0.012). Recurrence-free survival rates at 1, 3, and 5 years was 68.9%, 28.8%, and 18.5%, respectively, in patients with tumor diameters of 15–20 mm, versus 77.9%, 47.5%, and 29.9%, respectively, in patients with tumor diameters of less than 15 mm (Fig 4).

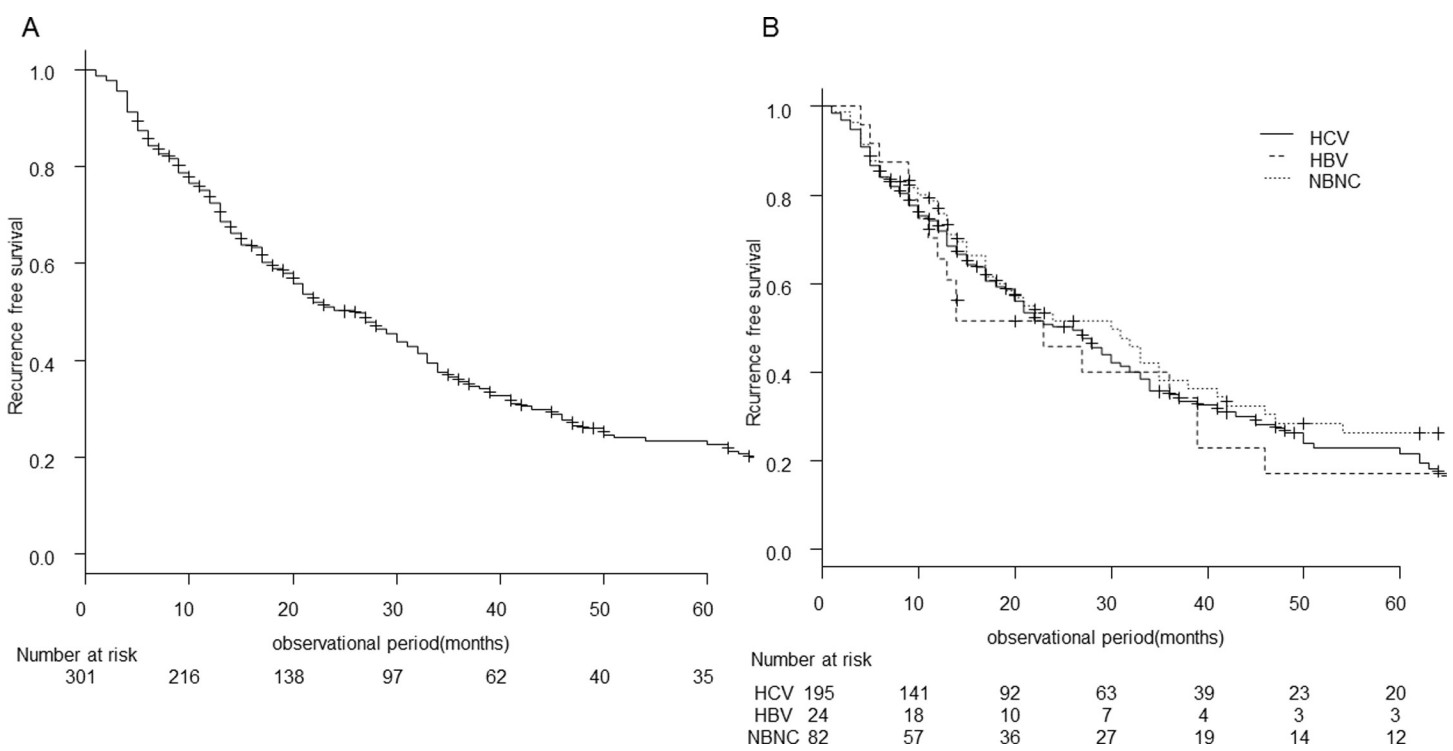

**Fig 2. Recurrence-free survival of very-early-stage HCC undergoing RFA.** The recurrence-free survival curve for all patients before (A) and after stratification by background liver disease (B).

In patients with active HCV infection during the observation period (HCV non-SVR), the median recurrence-free survival was 20 (95% CI 15–26) months compared to 31 (95% CI 23–38) months in the others (p < 0.001). Recurrence-free survival rates at 1, 3, and 5 years was 66.0%, 28.9%, and 16.3%, respectively, in the HCV non-SVR group, versus 77.6%, 41.6%, and 28.2% in the others (Fig 5).

## The impact of the timing of HCV eradication on overall survival and recurrence-free survival

Furthermore, when we compared the overall survival and recurrence-free survival among patients in the HCV SVR group in terms of the timing of achieving SVR (Fig 6), overall survival was not significantly different between patients who achieved SVR before HCC and those who achieved SVR after RFA treatment for HCC (p = 0.11) (Fig 6A). Recurrence-free survival did not also differ significantly between the two groups. The median recurrence-free survival was 29 (95% CI, 18–NA) months in patients who achieved SVR before HCC compared to 41 (95% CI, 29–74) months in those who achieved SVR after RFA treatment for HCC (p = 0.49) (Fig 6B).

The ALBI scores at the time of treatment, 1, 3, and 5 years were -2.56±0.38, -2.67±0.38, -2.83±0.42, and -2.82±0.51 (p = 0.006), respectively, and liver function showed a trend toward improvement. On the other hand, in the HCV non-SVR group, the ALBI scores at the time of treatment, 1, 3, and 5 years were -2.29±0.49, -2.21±0.51, -2.21±0.56, and -2.13±0.59 (p = 0.3), respectively, and liver function did not change significantly.

**Table 2. Factors related to overall survival on Cox proportional hazards analysis.**

| Variable | | Univariate analysis | | | Multivariate analysis | | |
|---|---|---|---|---|---|---|---|
| | | HR | 95%CI | P value | HR | 95%CI | P value |
| Age(years old) | ≦ 69 | 1 | | | | | |
| | ≧ 70 | 1.45 | 0.96–2.20 | 0.075 | | | |
| Sex | female | 1 | | | | | |
| | male | 1.11 | 0.74–1.64 | 0.62 | | | |
| AST(IU/L) | ≦ 40 | 1 | | | | | |
| | ≧ 41 | 2.23 | 1.4–3.56 | <0.001 | 1.51 | 0.75–3.02 | 0.25 |
| ALT(IU/L) | ≦ 40 | 1 | | | | | |
| | ≧ 41 | 1.57 | 1.04–2.37 | 0.031 | 0.94 | 0.54–1.64 | 0.82 |
| PLT($10^4$/μL) | ≦ 9.9 | 1 | | | | | |
| | ≧ 10 | 0.64 | 0.42–0.97 | 0.035 | 0.87 | 0.54–1.41 | 0.58 |
| PT(%) | ≦ 79.9 | 1 | | | | | |
| | ≧ 80 | 1.12 | 0.65–2.10 | 0.61 | | | |
| mALBI grade | 1 or 2a | 1 | | | | | |
| | 2b or 3 | 2.02 | 1.35–3.01 | <0.001 | 1.44 | 0.87–2.38 | 0.16 |
| AFP(ng/mL) | ≦ 4.9 | 1 | | | | | |
| | ≧ 5 | 3.02 | 1.32–6.92 | 0.009 | 1.21 | 0.47–3.12 | 0.69 |
| DCP(mAU / mL) | ≦ 39.9 | 1 | | | | | |
| | ≧ 40 | 1.42 | 0.88–2.28 | 0.15 | | | |
| Tumor size(mm) | ≦ 14 | 1 | | | | | |
| | ≧ 15 | 1.07 | 0.71–1.60 | 0.77 | | | |
| etiology | | | | | | | |
| HBs Ag positive | no | 1 | | | | | |
| | yes | 0.9 | 0.45–1.80 | 0.77 | | | |
| HCV Ab positive and not achived SVR | no | 1 | | | | | |
| | yes | 2.91 | 1.89–4.45 | <0.001 | 2.17 | 1.31–3.60 | 0.003 |
| both negative | no | 1 | | | | | |
| | yes | 0.61 | 0.37–1.01 | 0.057 | | | |

### The impact of entry period on survival

Comparing the cases before and after 2011 (the year where DAA became available), there was no statistically significant difference in OS and RFS of HBV or NBNC patients. Likewise, comparing the cases before and after 2009 (the year where sorafenib became available), there was no statistically significant difference in OS and RFS of HBV or NBNC patients. Therefore, the impact of the inclusion period of the patients or the effect of post-treatment of the patients seems to be minimal in this study cohort.

### Discussion

RFA is an effective treatment option for early-stage HCC because of its good long-term local control [19] and low frequency of complications [23]. Previous reports on the prognosis of patients treated using RFA for very-early-stage HCC showed that the 3- and 5-year survival rates were 80.3–88.6% and 72.0–76.0%, respectively, and the recurrence-free survival rates were 68.8–76.3%, 38.0–48.2%, and 24.0–29.3% at 1, 3, and 5 years, respectively [24–28]. On the other hand, none of them showed long-term survival beyond five years. In our study, the 3-year and 5-year survival rates were similar to those in previous reports. We further reported the survival rates at 7 and 10 years as 51.1% and 33.2%, respectively. Also, the present study

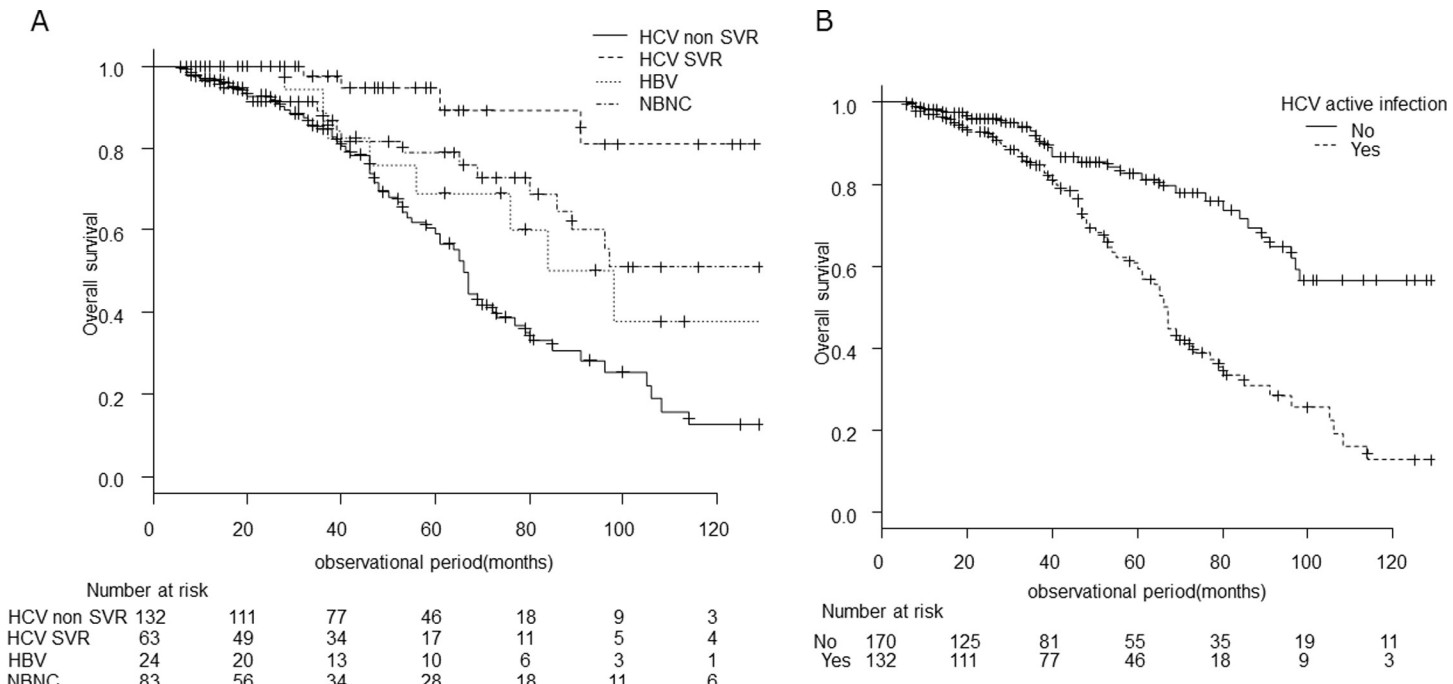

**Fig 3. Overall survival according to the presence of active HCV infection.** The overall survival curve after stratification by background liver disease, dividing patients with HCV into patients with SVR and active HCV infection during the observation period (HCV non-SVR) (A). Patients with active HCV infection and others, including HBV, HCV SVR, and NBNC were also compared (B). Patients with active HCV infection had a significantly shorter overall survival than others (p < 0.001).

elucidated the negative impact of active HCV infection on the prognosis. Excluding patients with active HCV infection throughout the observation period, the survival rates were 91.0%, 82.8%, 71.5%, and 56.5% at 3, 5, 7, and 10 years, respectively. These results show the benefit of HCV eradication in early-stage HCC with the background of HCV infection.

As for antiviral therapy after curative treatment of HCC, IFN treatment has been reported to improve overall survival and recurrence-free survival [29]. However, the low rate of SVR and the high frequency of adverse effects were problems facing its application. One RCT in HBV patients reported that adefovir after hepatectomy reduced the risk of late recurrence after two years of treatment and prolonged survival [30]. As for the HCV, SVR can now be achieved with a high rate with only a few side effects with the advent of DAA therapy [10–13]. Several observational studies have reported that HCV treatment generally reduces or improves liver function deterioration [31,32] and contributes to the improvement of overall survival by reducing the risk of non-liver diseases such as cardiovascular disease and neurological diseases [33–35].

On the other hand, whether or not it contributes to the improvement of recurrence-free survival is debatable. Initially, the possibility of early recurrence after DAA treatment was suggested [36,37]; however, subsequent reports, including meta-analyses, have shown no difference in the risk of recurrence when compared to IFN treatment [38–41]. Although time bias cannot be excluded completely, a recent report showed that the recurrence rate was significantly lower in the group that achieved SVR by DAA treatment after HCC treatment compared with the untreated group [32,42]. Furthermore, that achieving SVR reportedly improves overall survival and recurrence-free survival regardless of whether the patient is treated before or after HCC [43]. Under these backgrounds, our study focused on very early-stage HCC treated curatively using RFA. It confirmed that active HCV infection is a risk factor for early

**Table 3. Factors related to recurrence on Cox proportional hazards analysis.**

| Variable | | Univariate analysis | | | Multivariate analysis | | |
|---|---|---|---|---|---|---|---|
| | | HR | 95%CI | P value | HR | 95%CI | P value |
| Age(years old) | ≦ 69 | 1 | | | | | |
| | ≧ 70 | 1.12 | 0.84–1.48 | 0.45 | | | |
| Sex | female | 1 | | | | | |
| | male | 1.22 | 0.92–1.61 | 0.17 | | | |
| AST(IU/L) | ≦ 40 | 1 | | | | | |
| | ≧ 41 | 1.43 | 1.07–1.92 | 0.014 | 1.12 | 0.77–1.62 | 0.56 |
| ALT(IU/L) | ≦ 40 | 1 | | | | | |
| | ≧ 41 | 1.15 | 0.87–1.53 | 0.33 | | | |
| PLT($10^4$/μL) | ≦ 9.9 | 1 | | | | | |
| | ≧ 10 | 0.91 | 0.68–1.22 | 0.54 | | | |
| PT(%) | ≧ 80 | 1 | | | | | |
| | ≦ 79.9 | 1.14 | 0.77–1.67 | 0.51 | | | |
| mALBI grade | 1 or 2a | 1 | | | | | |
| | 2b or 3 | 1.32 | 0.99–1.76 | 0.054 | | | |
| AFP(ng/mL) | ≦ 4.9 | 1 | | | | | |
| | ≧ 5 | 1.7 | 1.14–2.54 | 0.01 | 1.27 | 0.80–2.02 | 0.31 |
| DCP(mAU / mL) | ≦ 39.9 | 1 | | | | | |
| | ≧ 40 | 1.35 | 0.95–1.93 | 0.095 | | | |
| Tumor size(mm) | ≦ 14 | 1 | | | | | |
| | ≧ 15 | 1.44 | 1.08–1.93 | 0.015 | 1.5 | 1.10–2.01 | 0.011 |
| etiology | | | | | | | |
| HBs Ag positive | no | 1 | | | | | |
| | yes | 1.08 | 0.65–1.77 | 0.77 | | | |
| HCV Ab positive and not achived SVR | no | 1 | | | | | |
| | yes | 1.61 | 1.22–2.12 | <0.001 | 1.47 | 1.06–2.05 | 0.022 |
| both negative | no | 1 | | | | | |
| | yes | 0.82 | 0.59–1.14 | 0.24 | | | |

recurrence of HCC and shorter survival. One of the suspected mechanisms for the improvement of survival in HCV-HCC patients with SVR is the improvement of liver function as described in the present study. These results suggest that antiviral treatment is desirable for very early-stage HCC with hepatitis C to prolong survival.

Regarding the comparison between RFA and hepatic resection in very early-stage HCC of BCLC stage 0, most previously reported retrospective studies show that RFA and hepatic resection are equivalent in terms of overall survival and that hepatic resection is superior in terms of recurrence-free survival. However, there is no established consensus [24–28]. There are no RCTs of BCLC stage 0 only, and small RCTs of early-stage HCC up to BCLC stage 0 or A are reported. The prognosis of overall survival and recurrence-free survival for RFA and hepatectomy are equivalent [44,45], while some studies show that hepatectomy has a better prognosis for overall survival and recurrence-free survival [46]. The SURF study results, a recent RCT on HCC within the Milan criteria in Japan, showed that recurrence-free survival of RFA and hepatic resection were equivalent [18], and we believe that RFA has advantages in terms of hospital stay and invasiveness. Besides, no-touch ablation using a bipolar RFA system is expected to improve larger tumors' outcomes [47,48]. The US-US overlay fusion guidance can be highly effective for safety margin achievement in RFA [49]. RFA treatment is also reportedly effective for some intermediate-stage tumors [50].

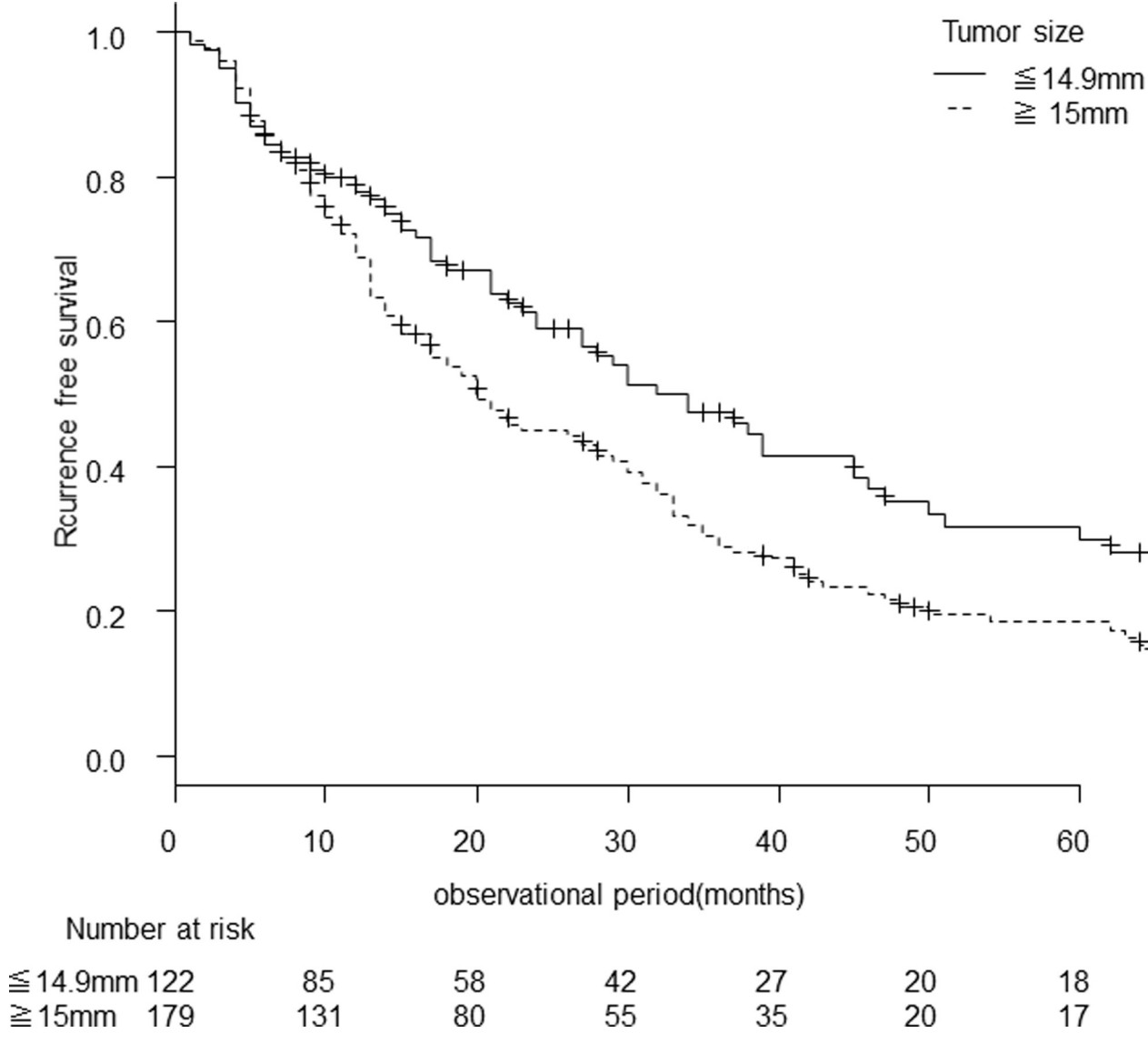

**Fig 4. Recurrence-free survival stratified by size of HCC.** The recurrence-free survival curve for two risk factors; the tumor diameter and the presence of active HCV infection. The recurrence-free survival was shorter in patients with larger tumors (p = 0.012).

This study has several limitations. First, most of the target lesions are diagnosed by CT or MRI images, and histological evaluation was not available. Histological differentiation and vascular invasion are considered prognostic factors and were not evaluated in this study. However, in this study, very early-stage HCC with a single lesion measuring less than 20 mm was targeted. It was difficult to collect tumor tissue in all patients before treatment.

Second, it was a single-center, retrospective study that did not consider the concomitant bias, including the time bias of when SVR was achieved. Also, the devices used differed depending on the time of RFA implementation, and we were not able to examine differences by the device.

In conclusion, the long-term prognosis of very early-stage HCC (BCLC stage 0) was favorable. Active HCV infection had a negative impact on survival and recurrence, confirming the benefit of achieving SVR by antiviral therapy in hepatitis C virus-infected patients.

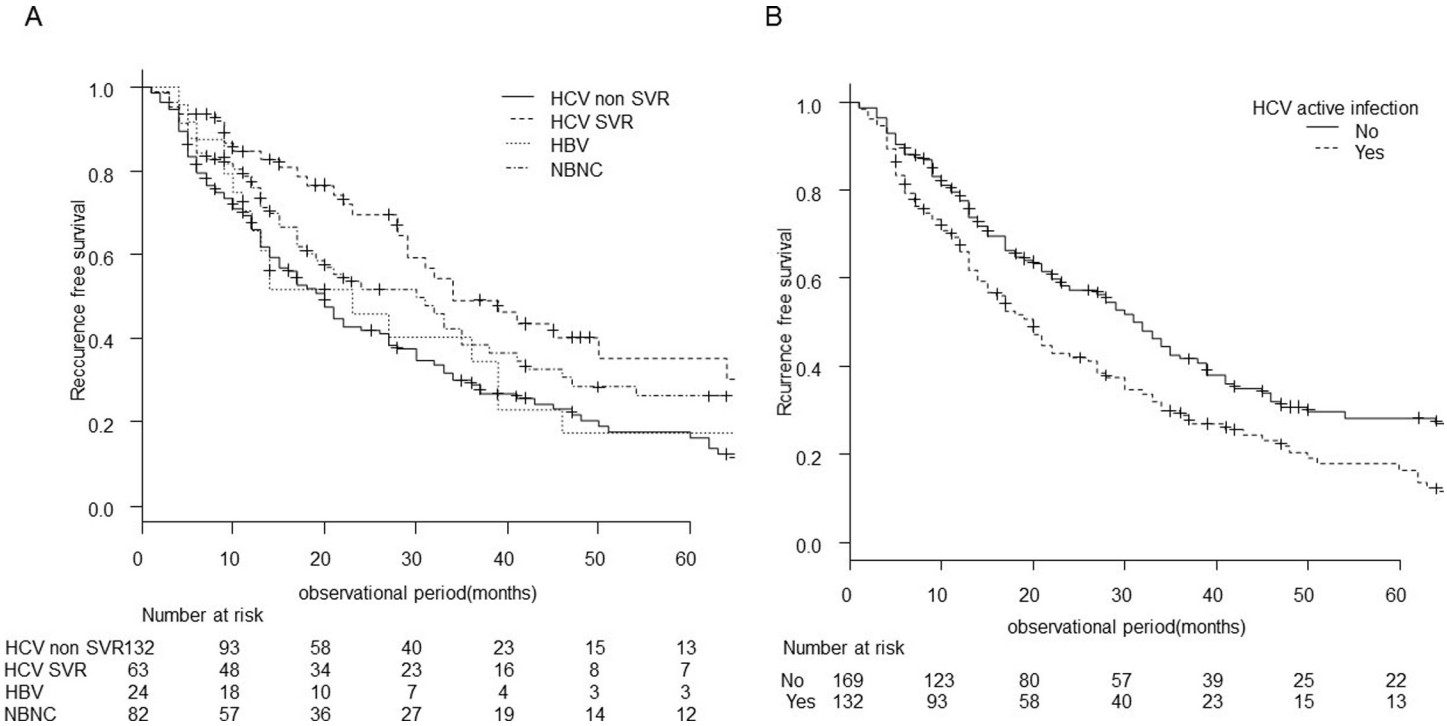

**Fig 5. Recurrence-free survival stratified by the presence of active HCV infection.** The recurrence-free survival curve after stratification by background liver disease, dividing patients with HCV into patients with SVR and active HCV infection during the observation period (HCV non-SVR) (A). Patients with active HCV infection and others, including HBV, HCV SVR, and NBNC were also compared (B). The recurrence-free survival was shorter in patients with active HCV infection (A) (p<0.001).

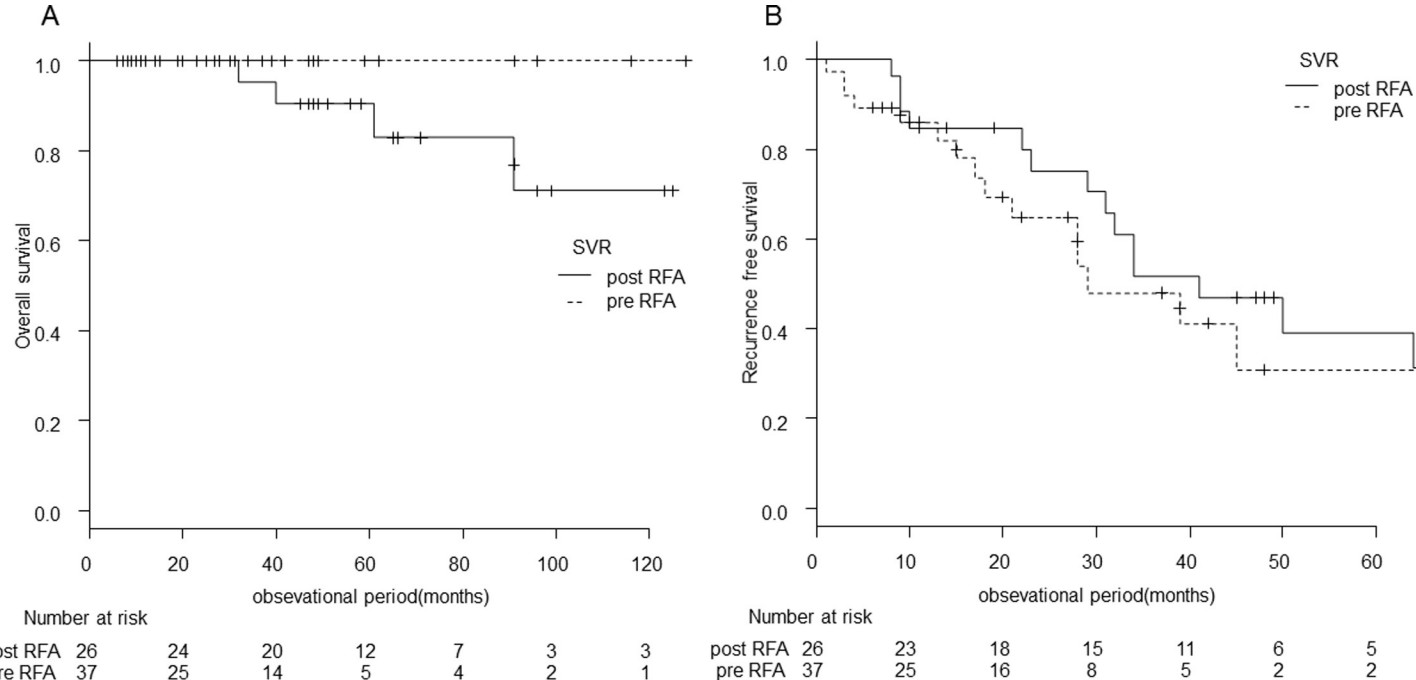

**Fig 6. Overall survival and recurrence-free survival according to the timing of the achievement of HCV SVR.** The overall survival and recurrence-free survival in the HCV SVR group in terms of the timing the achievement of SVR. The overall survival and the recurrence-free survival did not differ significantly between patients who achieved SVR before HCC and those who achieved SVR after RFA treatment for HCC; (p = 0.11) (A) and (p = 0.49) (B), respectively.

## Author Contributions

**Data curation:** Kenta Takaura, Kento Inada, Sakura Kirino, Kouji Yamashita, Tomohiro Muto, Leona Osawa, Shuhei Sekiguchi, Yuka Hayakawa, Mayu Higuchi, Shun Kaneko, Chiaki Maeyashiki, Nobuharu Tamaki, Yutaka Yasui, Jun Itakura, Kaoru Tsuchiya, Hiroyuki Nakanishi, Yuka Takahashi.

**Investigation:** Kenta Takaura.

**Methodology:** Masayuki Kurosaki.

**Supervision:** Masayuki Kurosaki, Namiki Izumi.

**Validation:** Masayuki Kurosaki.

**Writing – original draft:** Kenta Takaura.

**Writing – review & editing:** Masayuki Kurosaki, Namiki Izumi.

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
