## [Decision Letter · Decision Letter 0]

7 Oct 2021

PONE-D-21-26827The impact of background liver disease on the long-term prognosis of very-early-stage HCC after ablation therapyPLOS ONE

Dear Prof. Namiki Izumi,

Thank you for submitting your manuscript to PLOS ONE. After careful consideration, we feel that it has merit but does not fully meet PLOS ONE’s publication criteria as it currently stands. Therefore, we invite you to submit a revised version of the manuscript that addresses the points raised during the review process.

We look forward to receiving your revised manuscript.

Kind regards,

Tatsuo Kanda, M.D., Ph.D.

Academic Editor

PLOS ONE

Journal Requirements:

"I have read the journal's policy and the authors of this manuscript have the following competing interests: Namiki Izumi, Masayuki Kurosaki, and Kaoru Tsuchiya received lecture fees from Eisai, Bayer, Lilly, and Chugai.This does not alter our adherence to PLOS ONE policies on sharing data and materials."

We note that you received funding from a commercial source: Eisai, Bayer, Lilly, and Chugai

Reviewers' comments:

Reviewer's Responses to Questions

**Comments to the Author**

1. Is the manuscript technically sound, and do the data support the conclusions?

Reviewer #1: Yes

Reviewer #2: Yes

2. Has the statistical analysis been performed appropriately and rigorously? 

Reviewer #1: Yes

Reviewer #2: Yes

3. Have the authors made all data underlying the findings in their manuscript fully available?

Reviewer #1: Yes

Reviewer #2: Yes

4. Is the manuscript presented in an intelligible fashion and written in standard English?

Reviewer #1: Yes

Reviewer #2: Yes

5. Review Comments to the Author

Reviewer #1: The manuscript submitted by Takaura et al. describes the long-term prognosis of hepatocellular carcinoma (HCC) treated with radiofrequency ablation (RFA) at a very early stage (the Barcelona Clinical Liver Cancer classification stage 0). Specifically, Achieving sustained virological response (SVR) in hepatitis C was essential for further prognosis improvement. I think this manuscript still has some problems, as indicated below.

Methods

Are there any exclusion criteria?

The authors should describe the detail of the RFA procedure, such as approach, device, or protocol.

The ethics approval number should be clearly stated.

I could not find anything about informed consent.

Results

The authors should describe the success rate, frequency of complications, and degree of safety margin for RFA.

In Table 1, the range of tumor diameter is listed as 8-20 mm. This study focused on very early-stage HCC with a single lesion measuring less than 20 mm.

Was there any effect of post-treatment on overall survival (OS)?

Discussion

In general, liver function is an important prognostic factor for HCC. The authors should consider that modified albumin-bilirubin was not extracted as an independent factor contributing to OS.

In this study, was there a statistically significant difference in the changes in liver function between the HCV SVR group and the HCV non-SVR group?

Reviewer #2: In this manuscript, authors investigated the impact of the background liver disease on long-term prognosis in the BCLC stage 0 HCC treated with RFA, with a particular focus on the role of active HCV infection.

They report that active HCV infection was an independent factor influencing the prognosis of patients with the early small HCC treated with RFA.

This study is interesting and addresses an important issue. On the other hand, there are some concerns. an important issue. On the other hand, there are some concerns.

1) The study period for the inclusion of patients is quite long, from 1999 to 2010. Since the advancement of medicine in this long period, OS and RFS may differ between patients who were included earlier and those who were included more recently, and such differences may affect the study results. Therefore, authors better show and/or describe the impact of the inclusion period of the patients.

2) In relation to 1), in the era of IFN therapies, many patients could not achieve SVR. If so, is it possible that most of the HCC patients with active HCV infection participated in the early days of this study?

3) What is a suspected mechanism for the improvement of survival in HCV-HCC patients with SVR ? Did anti-HCV therapy decrease the HCC recurrence rate? Or did anti-HCV therapy have an impact on the maintaining liver functional reserve in HCC therapy? The authors should compare the clinical course of liver function in those with and without SVR.

4) In Figure 3 and 4, the OS and RFS of HBV, NBNC and HCV-SVR also should be separately demonstrated.

6. PLOS authors have the option to publish the peer review history of their article (what does this mean?). If published, this will include your full peer review and any attached files.

Reviewer #1: No

Reviewer #2: No

---

## [Author Response · Author response to Decision Letter 0]

22 Dec 2021

Editorial office

Plos One

Dear editor:

Thank you very much for your kind review of our manuscript. Taking into account all the suggestions and criticism raised by the editors and reviewers, we have now revised our manuscript, and we now submit a revised manuscript for the consideration for the publication in Plos One. 

A point-by-point reply to the reviewers’ comments is attached on a separate document. We believe that we have successfully answered all the suggestions and criticisms and that our manuscript is now suitable for publication. We look forward to hearing from you.

The corresponding author warrants that the article is original, is not under consideration by another journal, and has not been previously published. All authors have seen and approved the manuscript, and they have contributed significantly to this work. 

I have read the journal's policy and the authors of this manuscript have the following competing interests: Namiki Izumi, Masayuki Kurosaki, and Kaoru Tsuchiya received lecture fees for speaker’s bureau from Eisai, Bayer, Lilly, and Chugai. Since these lectures were about systemic therapy for unresectable HCC, the content was not related at all with the present study. As a natural consequence, this does not alter our adherence to PLOS ONE policies on sharing data and materials. None of the authors have any other relevant declarations relating to employment, consultancy, patents, products in development, marketed products.

I sign for and accept responsibility for this material on behalf of any and all co-authors. 

I look forward to hearing from you.

Sincerely yours,

Namiki Izumi

Division of Gastroenterology and Hepatology

Musashino Red Cross Hospital 

1-26-1 Kyonan-cho, Musashino-shi, Tokyo, 180-8610, Japan

Tel: +81-422-32-3111 

Fax: +81-422-32-9551

E-mail address: izumi012@musashino.jrc.or.jp

---

## [Decision Letter · Decision Letter 1]

3 Feb 2022

The impact of background liver disease on the long-term prognosis of very-early-stage HCC after ablation therapy

PONE-D-21-26827R1

Dear Prof. Namiki Izumi,

We’re pleased to inform you that your manuscript has been judged scientifically suitable for publication and will be formally accepted for publication once it meets all outstanding technical requirements.

Kind regards,

Tatsuo Kanda, M.D., Ph.D.

Academic Editor

PLOS ONE

Additional Editor Comments (optional): Authors reported that the long-term prognosis of very early-stage HCC (BCLC stage 0) was favorable and that the importance of sustained virological responce is very importnat for the prognosis of hepatitis C virus associated-HCC at very early stage and RFA-treatment. Thank you for giving me the opportunity of reviewing this wonderful manuscript.

Reviewers' comments:

Reviewer's Responses to Questions

**Comments to the Author**

1. If the authors have adequately addressed your comments raised in a previous round of review and you feel that this manuscript is now acceptable for publication, you may indicate that here to bypass the “Comments to the Author” section, enter your conflict of interest statement in the “Confidential to Editor” section, and submit your "Accept" recommendation.

Reviewer #1: All comments have been addressed

Reviewer #2: All comments have been addressed

2. Is the manuscript technically sound, and do the data support the conclusions?

Reviewer #1: Yes

Reviewer #2: Yes

3. Has the statistical analysis been performed appropriately and rigorously? 

Reviewer #1: Yes

Reviewer #2: Yes

4. Have the authors made all data underlying the findings in their manuscript fully available?

Reviewer #1: Yes

Reviewer #2: Yes

5. Is the manuscript presented in an intelligible fashion and written in standard English?

Reviewer #1: Yes

Reviewer #2: Yes

6. Review Comments to the Author

Reviewer #1: The manuscript submitted by Takaura et al. describes the long-term prognosis of hepatocellular carcinoma (HCC) treated with radiofrequency ablation (RFA) at a very early stage (the Barcelona Clinical Liver Cancer classification stage 0). The authors correctly understood the points raised and made the appropriate corrections. This paper is a significant contribution.

Reviewer #2: All concerns have been adequately addressed and corrected in the revised manuscript.

Therefore, the manuscript is now considered to be suitable for publication.

7. PLOS authors have the option to publish the peer review history of their article (what does this mean?). If published, this will include your full peer review and any attached files.

Reviewer #1: No

Reviewer #2: No

---

## [Editor Report · Acceptance letter]

8 Feb 2022

PONE-D-21-26827R1 

The impact of background liver disease on the long-term prognosis of very-early-stage HCC after ablation therapy 

Dear Dr. Izumi:

I'm pleased to inform you that your manuscript has been deemed suitable for publication in PLOS ONE. Congratulations! Your manuscript is now with our production department. 

Kind regards, 

on behalf of

Dr. Tatsuo Kanda 

Academic Editor

PLOS ONE